# Tribological and Thermal Transport Performance of SiO$_2$-Based Natural Lubricants

**Jaime Taha-Tijerina [1],*** , **Karla Aviña [1] and Jose Manuel Diabb [2]**

[1] Engineering Department, Universidad de Monterrey, Av. Morones Prieto 4500 Pte., San Pedro Garza García 66238, Mexico

[2] Facultad de Ingeniería Mecánica y Eléctrica, Universidad Autónoma de Nuevo León, Av. Universidad S/N, San Nicolás de los Garza 66450, Mexico

* Correspondence: jose.taha@udem.edu; Tel.: +52-1-811-570-7510

**Abstract:** Fluids and lubricants are critical for the mechanical manufacturing processing of metals, due to a high amount of friction generated, also reflected as heat, could wear and damage tooling and machine components. The proper application of lubricants increases machinery lifetime, decreases long-term costs, and energy and time consumption due to the maintenance or components exchange/repairs. Besides being non-renewable, mineral oils bring consequences to the environment due to their low biodegradability and could affect the user with respiratory and skin diseases. Recently, due to an increase in environmental awareness, the search of biocompatible and efficient lubricants has become a technology goal. The vegetable oil-based lubricants are slowly emerging as ecofriendly and high-performance alternatives to petroleum-based lubricants. This study evaluates soybean, sunflower, corn and paraffinic oils reinforced with SiO$_2$ nanoparticles. The thermal and tribological evaluations were performed varying the temperature and nanofiller concentrations. The thermal conductivity improvements were observed for all nanolubricants as the temperature and filler fraction increased. The highest thermal conductivities were observed at 323 K with 0.25 wt % SiO$_2$. The soybean and corn oils unveiled a maximum enhancement of ~11%. The tribological evaluations showed that SiO$_2$ addition, even in small concentration, resulted into a significant improvement on a load-carrying capacity. For instance, at 0.25 wt % enhancements of 45% and 60% were observed for soybean and sunflower oils, respectively. The coefficient of friction performance also showed enhancements between 10% and 26%.

**Keywords:** lubricant additives; thermal transport; wear testing; load-carrying capacity; SiO$_2$

## 1. Introduction

In the manufacturing industry, it is well-known that metal-working fluids improve the efficiency of machining in terms of increased tool life, improved tolerance and surface finish and reduced cutting force or vibrations [1]. Moreover, the heat produced while machining affects the sharpness and hardness of the cutting tools, which result in their early damage or failure. Hence, it is important to apply the proper cutting fluid or lubricant to overcome these issues.

The current manufacturing processes involve metal sheet forming, stamping, rolling, machining, etc. The process itself generates a high amount of friction, which could wear and damage the machines tooling and/or components that are manufactured. Therefore, this friction must be solved with the aid of lubricants, coatings or the combination of both. Currently, the most common tool to counterattack friction and heat generated by a system, are lubricants. They generate a very thin film between two surfaces, which helps to increase the useful life of the machinery and reduce the damage of the component in its manufacture [1]. Due to the high pressures, elevated temperature, among other

factors, the lubricants deteriorate very fast which causes a problem because the lubricants must be replaced very often, and the cost to the industries increases. One critical aspect in industry is that the production lines must stop often for the lubricants to be replaced, which represents an expensive practice that companies try to minimize. The additives are an alternative to increase the lubricants' physical or mechanical performance (lifetime). They can also help to increase the viscosity, helping to improve the lubricant's properties.

Heat dissipation is a significant phenomenon in industrial processes and thermal systems. Together with the design and devices miniaturization, advanced fluids for the enhancement of heat dissipation efficiency have been required for global competition and the environmental challenges. Hence, the efforts on the design, development and research of thermal systems and technologies for the improvement of thermal properties of working lubricants and coolants for the enhancement of heat transfer efficiency are crucial [2].

On a specific operation process such as machining, the generated heat moves into the cutting tool, workpiece, and the surrounding atmosphere by conduction, convection, or radiation, depending upon the ambient conditions [2]. The cutting fluids and lubricants play a paramount role by cooling and maintaining a suitable working piece interface, washing away debris and chips from machining areas. Nevertheless, this conventional way of lubricity and cooling serves the purpose up to an extent. The excessive use of certain cutting fluids could pollute the environment and may even be hazardous for human operators, due to their lack of ecological balance owing to their non-biodegradable or toxic characteristics.

The lubricants demand, on a worldwide basis, move forward by more than 10% year-on-year to just approximately 32 million metric tons in 2009, growing to nearly reach as 36 million metric tons level in 2015 [3]. An important fact to be taken into account is that approximately 85% of lubricants being used around the world are petroleum-based oils [4]. As it is evident, the environmental issues related to the massive usage of petroleum-based fluids and the geopolitical strategies concerning crude oil manipulation are considered main drivers behind the introduction of alternative lubricants and fuels from renewable raw materials [5–7]. Further, the critical negative impact on the environmental and health-related consequences, i.e., skin and respiratory diseases, must be taken into account. Thus, to reduce the bulk usage of conventional cutting fluids and to minimize their adverse effects on operators and the environment, alternatives to these fluids are being explored in detail by the research community.

A slow but steady move towards the use of eco-friendly or more readily biodegradable fluids has taken place during recent years. Biodegradability has become one of the most important design parameters both in the selection of base fluids and in the overall formulation of the finished lubricant. In this sense, natural or vegetable fluids arose in the past decades. Vegetable fluids are renewable resources, eco-friendly non-toxic materials which pose no workplace health hazards, possess excellent lubricity, high load-carrying capacity, high thermal conductivity, a very high viscosity index, high flash points and are biodegradable, as compared to mineral oil-based working fluids.

Despite their superb properties, they also present certain disadvantages, which impact greatly on their performance, for instance, vegetable fluids in their natural form have limited applications as industrial working materials due to their poor thermal or oxidation stability [8–14], low temperature performance [15–17] and other tribo-chemical degrading processes [18–20] that occur under severe conditions of stress, pressure, temperature, environment and metal substrates. Vegetable fluids have limited oxidative stability, which have been studied before and have reflected on their tribological performance [21]. The oxidation process results in deteriorating the quality of the oil which in turn may have a high influence on the final product quality and shelf life of the vegetable oil-based lubricants [22–25]. For instance, low oxidative stability of the oil results in faster oxidizing rate, becoming thick and polymerizing to a plastic-like consistency [22]. Furthermore, due to their chemistry, the vegetable fluids' structure strongly interacts with metallic surfaces, resulting in reduced friction and wear, forming a high strength lubricant film. These strong intermolecular interactions are also resilient to changes in temperature resulting in high viscosity [21].

Recently, many investigations have been carried out on the applications where nanoparticles reinforced conventional fluids and lubricants. Cooling and the reduction of wear and friction are critical technical challenges facing different industries, which are dependent on the characteristics of nanoparticles such as the size, shape and filler fraction, among others [18,26–33].

Diverse vegetable fluids have been investigated for their thermal and tribological characteristics. Khedkar et al. [34] investigated the thermal performance of $Fe_3O_4$ nanoparticles within paraffin fluid on filler concentrations ranging from 0.01–0.10 vol %. The thermal conductivity improvement of ~20% for paraffin nanofluids at 0.10 vol % was observed by comparing them to a base fluid at room temperature. The thermophysical behavior of $TiO_2$ nanoparticles dispersed within various mixtures of bio-glycol and water (20:80% and 30:70% by volume) were studied by Sidik et al. [35]. The experiments conducted over temperature ranges from 30 and 80 °C, and at 0.5 and 2.0 vol % were performed. Thermal conductivity increased by increasing the temperature and filler concentration. The highest improvement in thermal conductivity was ~13% at 2.0 vol % of $TiO_2$ for 20:80% mixture ratio at 80 °C. The experiments on Cu-Zn 0.1–0.5% and the diverse types of conventional fluids (such as vegetable and paraffin oils) were performed by Kumar et al. [36] to study the rheological and thermal performance. All the samples displayed good stability for approximately 72 h. Furthermore, the hybrid nanofluids with vegetable oil showed marginally less stability. Kumar et al. concluded that based on the integrated thermal conductivity and viscosity performance, vegetable oil-based nanofluid showed 53% improvement, as compared to the other studied fluids.

On the tribological aspect, Ozcelik et al. [37] evaluated metal-cutting fluids with sunflower oil, canola oil, and mineral oils. It was observed that canola oil displayed better tribological performance compared to sunflower oil-based fluids which was attributed to the variation in the lengths of carbon chains. Further, the higher carbon content and higher viscosity of canola oil promotes a better lubrication. The research by Kumar et al. [38] on coconut oil with extreme pressure (EP) additives in machining AISI 1040 steel was performed. It was observed that coconut oil reduced the feed force by 31%, thrust force by 28%, cutting force by 20%, cutting tool temperature by 7%, and tool flank wear by 34% compared to other fluids. Zhang et al. [39,40] investigated soy-based cutting fluid to evaluate its performance compared to petroleum-based fluids. The soy-based fluid performed similar to petroleum products in surface roughness and significantly better than the dry cutting. Also, the behavior in reducing tool wear was similar for the soy-based fluid and the petroleum-based fluid. Trajano et al. investigated the tribological behavior of sunflower and soybean biolubricants reinforced with CuO and ZnO nanoparticles. For soybean nanofluids at 0.5 wt % CuO and ZnO reinforcement, the coefficient of friction (COF) improved by 11% and 18%, respectively. On the other hand, for sunflower nanofluids at same filler fraction, the COF improved by 22% and 20%, for CuO and ZnO reinforcement, respectively. It was observed that when ZnO was added to the lubricant, the percentage of film formation reduced drastically, and soybean biolubricant with CuO did not maintain the film throughout the tribological evaluation. According to Trajano et al., this difference is due to the polarity of biolubricants. This may be because the sunflower polarity is higher than soybean polarity, resulting in higher adsorption on a metal surface [41]. According to Hutching [42], vegetable oil, naturally, contains molecular species with boundary lubrication properties (like oleic acid). It is important to consider that sunflower oil has high polarity which increases the adhesion to a metal surface.

Due to their anti-wear properties, the lubricants based on mineral oils have been used in the mechanical industry for all kinds of lubrication such as hydraulic systems, industrial gears, automotive engines, etc. However, mineral oils can be toxic if they contain higher amounts of elements such as sulphur, phosphorus and heavy metals, and that is the reason why they are being slowly replaced from the industry [43,44].

More often, biodegradable synthetic products are used in environmentally sensitive areas [5]. The increased environmental awareness is a primary driving force for the novel technological developments in this field [5]. Vegetable oils are an alternate to mineral oils, because they have non-toxic properties and they are renewable and environmentally friendly. It has been proven that

these oils reduce hydrocarbon and carbon monoxide emission levels, which is the reason why the use of these oils has been increased recently in the industry. Further, these oils have excellent biodegradability and they can be used to solve tribological issues. Despite the environmental benefits of vegetable oils, their properties do not satisfy all of the required properties [14,45].

Thermal fluids are crucial to the system to remove the excess of heat, which is important to the industries of electric engineering, automotive and the medical field. Some of the lubricants or commercial fluids have low thermal properties so that the addition of nanoparticles are a suitable alternative to improve them. The nanofluids are a dispersion of nanosized elements into a base fluid (water or oils). Some of the advantages of the nanofluids include the improvement of thermal properties by removing the excess heat generated by a system, and tribological properties by reducing wear and COF [46,47].

Due to the good tribological properties (anti wear and friction reduction), silica nanoparticles have been investigated as additives to lubricants. These nanoparticles have a good dispersion within commercial lubricants, tunable properties and composition in a very predictable manner to meet the needs of particular applications, providing the best viscosity according to experimental and mathematical model studies [45,46]. The $SiO_2$ nanoparticles have a good performance on the physical, chemical and electrical field. These characteristics make this nanoparticle very functional and is used to improve the properties of materials, lubricants, catalyst and biomedicine [46–50]. Silica-based nanoparticles are robust inorganic materials which possess high specific surface areas with high surface silanol concentrations, which makes it easy to connect the functional groups, such as amino and carboxyl, to the silica nanostructures surface for further biomolecules binding [50].

It is known that the nanoparticles are added to the lubricants to reducer wear and COF to the mineral oils, but there are a few reports that work with vegetable oils. It has been shown that vegetable oils have excellent lubrication and they could outperform mineral oils. The vegetable nanofluids were developed to improve the previous results. It has been proved that smaller particles are more likely to interact with the surfaces. This behavior helps the nanoparticles added to the base fluid interact with the roughness and reduce wear and COF [14,51]. Apart from that, another important aspect with this nanofluid based in vegetable oils is the life time. The vegetable oils have the restriction of the temperature. This is because they are composed of different fatty acid molecules. These molecules are presented as double bound, making the vegetable oil more prone to oxidize or degrade when it is exposed to a heat higher than 100 °C. To combat this problem, there are some studies that modify the chemical of these oils, or add some additives and property enhancers. The addition of antioxidants has considerable effectiveness. In addition, the degradation of vegetable oils cause several chemicals to be generated which are considered toxic and potentially carcinogenic [45,52–54].

In this work, the effect of the addition of the $SiO_2$ nanostructures homogeneously dispersed within vegetable fluids such as soybean oil, sunflower oil and corn oil, to improve their thermal and tribological properties namely thermal conductivity, COF, wear resistance and load-carrying capacity, are determined. The $SiO_2$ nanoparticles are selected due to their thermal conductance, small size and geometry, which promote the filling of valleys in the substrate and take advantage of the rolling effect. The research gives information on the behavior of this nanofluids, in their use to reduce wear, and to provide information on their thermal properties compared to mineral oil. Additionally, the study provides information on the viscosity of vegetable nanofluids.

## 2. Materials and Methods

### 2.1. Materials

In this work, a two-step method to homogeneously disperse the $SiO_2$ nanoparticles (size 10–20 nm, from SigmaAldrich, Toluca, México) within conventional fluids was used. This process assists in obtaining the stable homogeneous nanofluids by varying the filler concentration by weight, which

is further evaluated. The investigated fluids are soybean oil, sunflower oil, corn oil and paraffinic mineral oil, purchased from Sigma Aldrich. Table 1 shows the basic characteristics of these fluids.

**Table 1.** Material properties.

| Materials | Properties | | | |
|---|---|---|---|---|
| **Base Fluids** | **Density @ 20 °C (g/cm³)** | **Viscosity @ 24 °C (m·Pa·s)** | **Viscosity @ 40 °C (m·Pa·s)** | **Viscosity @ 100 °C (m·Pa·s)** |
| Soybean Oil | 0.9604 | 54.3 | 32.93 | 6.79 |
| Paraffinic Oil | 0.8900 | 37.8 | 24.0 | 4.80 |
| Corn Oil | 0.9100 | 52.3 | 30.8 | 6.57 |
| Sunflower Oil | 0.9197 | 68.0 | 40.05 | 8.65 |
| *Nanoparticles* | | | | |
| SiO₂ | Morphology: Spherical. Size: 10–20 nm | | | |

### 2.2. Nanofluids Preparation

The $SiO_2$ nanoparticles (described in Table 1) were homogeneously dispersed within the conventional fluids at different filler fractions: 0.05, 0.10, 0.15, 0.20 and 0.25 wt % for each conventional fluid. The extended water bath sonication (~3–4 h) was first used (Branson ultrasonic homogenizer model 5510 (Danbury, CT, USA, 40 kHz). The water bath temperature was maintained constant at room temperature (24 °C) to avoid possible nanoparticle agglomeration and fast sedimentation. Afterwards, a Metason 120T sonicator (Ballerup, Denmark) (output power of 70 W) was used for 1–2 h, according to a previous study by the group. The samples were maintained on a shelf for at least 3 weeks without significant sedimentation. The experimental evaluations were performed after 2 days of sample preparation.

## 3. Experimental Details

### 3.1. Thermal Experimentation

The thermal conductivity measurements at various filler fractions of $SiO_2$, ranging from 0.05 to 0.25 wt %, were carried out following the transient hot-wire (THW) technique. A Decagon Device Inc., model KD2 Pro (Pullman, WA, USA) with a KS-1 probe was used. This probe is calibrated using glycerol and the thermal conductivity value was verified up to 3 decimal points. For the temperature-dependent measurements, a thermal bath was used. The samples were thermally equilibrated for 10–15 min before each measurement. The thermal conductivities are the average of at least 8 consecutive measurements. The measured values were compared with the thermal conductivities of base fluids.

### 3.2. Tribological Experimentation

For the tribological evaluation, the load-carrying capacity at extreme pressures and the coefficient of friction (COF) were analyzed with a four-ball universal tribotester (balls material is an AISI 52100 steel, with a diameter of 12.7 mm, and 60 HRC), according to the ITEePib Polish method for testing lubricants under scuffing conditions [55]. This methodology was selected since it has shown to be more sensitive to extreme pressure additives [56–59], as well as being less time consuming. Table 2 shows the test parameters. This test determines the frictional torque until the seizure occurs at 10 N·m, when the oil film is destroyed. The maximum load at which the seizure occurs is called $P_{oz}$. In the case that the 10 N·m torque is not reached, the seizure load would be the maximum, 7200 N. The limiting pressure of seizure or $P_{oz}$, is calculated as follows:

$$P_{oz} = 0.52 \frac{P_{oz}}{WSD^2} \tag{1}$$

**Table 2.** The parameters of tests done using the universal ball tribotester.

| Parameters | ITEePib Polish Method |
| --- | --- |
| Time | 18 s |
| Velocity (RPM) | 500 |
| Temperature (°C) | 24 |
| Applied Force (N) | 0–7200 (linear increment) |

The wear scar diameter (WSD) is calculated by averaging the wear scar from the three stationary balls as measured by an optical microscope. The greater the $P_{oz}$ is, the best tribological characteristics the lubricant has.

## 4. Results and Discussion

### 4.1. Thermal Performance

Figure 1 depicts the temperature-dependent thermal conductivity evaluation performed to $SiO_2$ nanofluids at various filler fractions. The thermal conductivity of the investigated conventional fluids did not show significant temperature dependence (less than 2% at 50 °C, compared to room temperature). Furthermore, the thermal conductivity of the studied nanofluids was found to be gradually increased with the $SiO_2$ filler concentration increase. Furthermore, for all the nanofluids, the thermal conductivity was observed to improve as temperature also elevated, indicating the Brownian motion contribution in thermal transport behavior.

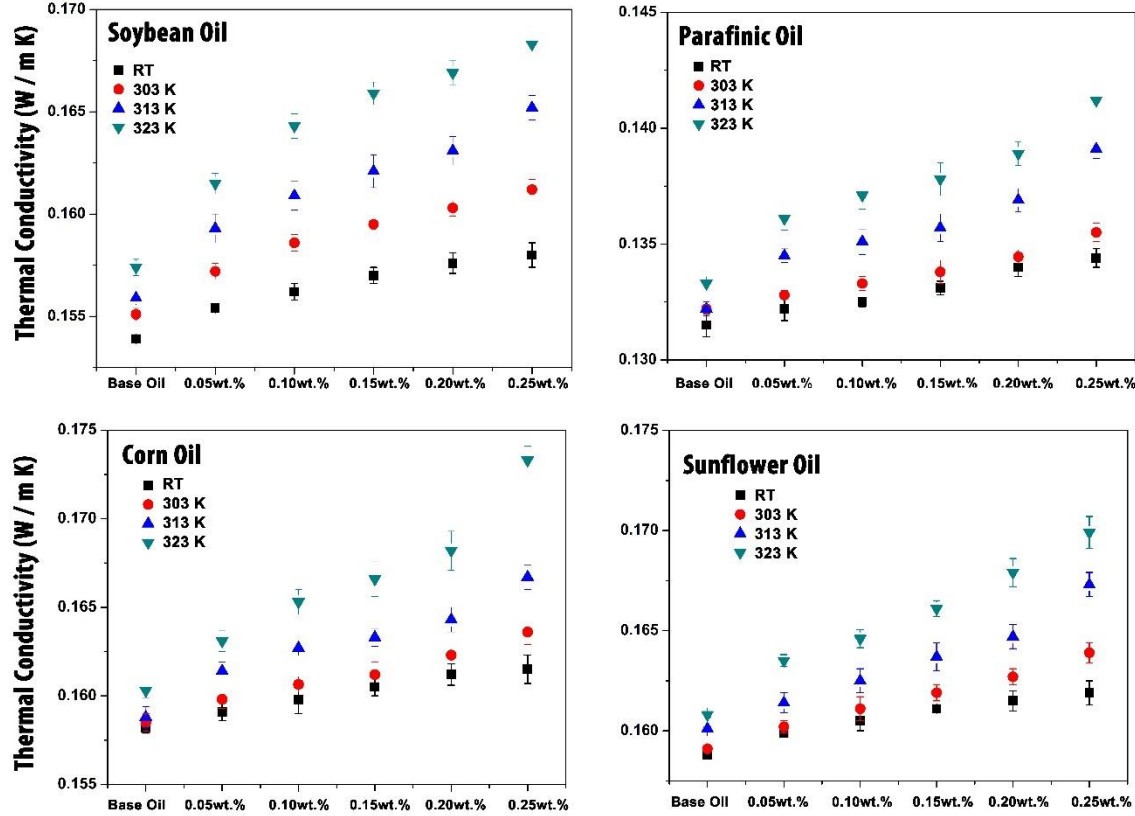

**Figure 1.** Thermal conductivity enhancement of various nanofluids under temperature-dependence evaluation (percentage of filler amount is mentioned).

The soybean nanofluid showed an increase on the thermal conductivity as the temperature and filler fraction increased. For instance, at 313 K, the improvements of 6% and 8% were obtained at 0.15

and 0.25 wt %, respectively. A maximum improvement of ~11% was observed at 0.25 wt % at 313 K. A similar improvement trend was observed for paraffinic oil, which thermal behavior reflected an enhancement of 5% and ~9% at 323 K, for 0.15 and 0.25 wt %, respectively, compared to base oil. Corn oil showed enhancement in thermal conductivity as the filler fraction increased as well. In this case, an improvement of 6% was obtained with 0.25 wt % at 313 K and a maximum enhancement of ~11% was observed for the 0.25 wt % concentration at 323 K. Furthermore, sunflower oil showed 6% and ~9% enhancement at 0.25 wt % at 313 and 323 K, respectively.

It is suggested that due to the low filler fraction of $SiO_2$, the observed enhancement on thermal conductivity is due to the interactions (collisions) between the oil molecules and the $SiO_2$ nanoparticles [60]. However, the temperature-dependent variations in thermal conductivity indicate that it is not just the percolation mechanism that increases the thermal conductivity, the Brownian motion also contributes to the thermal conductivity of $SiO_2$-based nanofluids. Moreover, for oil-based samples, liquid layering at the particle/liquid can also contribute to the enhancement in thermal transport behavior [61–66].

### 4.2. Tribological Performance

Figure 2 depicts the comparison between the $P_{oz}$ of various $SiO_2$ nanosystems. For instance, soybean oil showed an increase on the load-carrying capacity of ~18%, 28% and 45% at 0.05, 0.15 and 0.25 wt % filler fraction, respectively. A similar improvement trend was observed for paraffinic oil where the load-carrying capacity increased by 5%, 14% and ~20% at 0.05, 0.15 and 0.25 wt %, respectively. The effects of the $SiO_2$ nanoreinforcement on corn oil showed enhancements of 5%, 14% and ~25% at 0.05, 0.15 and 0.25 wt % filler fraction, respectively. The highest impact on the load-carrying capacity was achieved by sunflower oil, in which enhancements of 15%, 38% and ~60% were obtained for 0.05, 0.15 and 0.25 wt %, respectively.

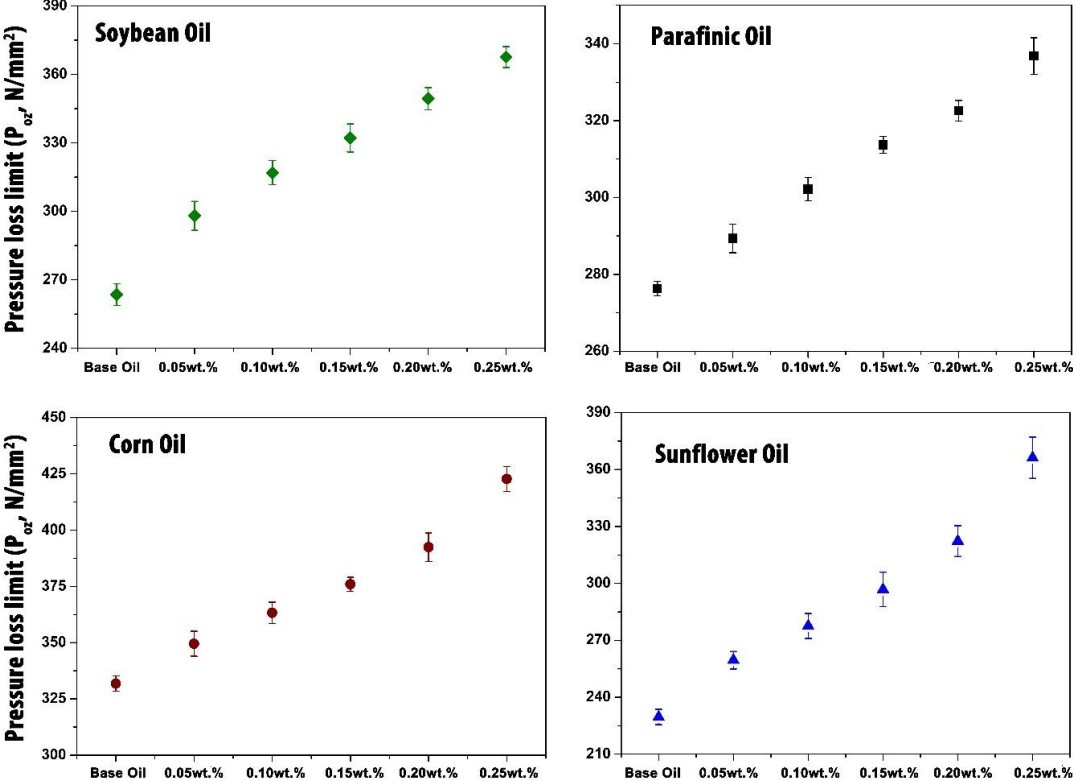

**Figure 2.** Tribological performance of various nanofluids under temperature-dependence evaluation (percentage of filler amount is mentioned).

The higher increase in the load-carrying capacity with the $SiO_2$ nanoreinforcements could be attributed to their tribosintering effect onto the worn surface, filling valleys and the shearing of trapped nanoparticles at the interface of contacting surfaces, thus making them smooth and lowering the frictional forces [67–70]. Similarly, the observed outstanding anti-wear performance can be associated with the hardness of the substrate and the formation of tribo-films as also observed by other authors [67–73].

Table 3 shows the effect of $SiO_2$ on COF during tribo-testings according to the ITEePib Polish method for testing lubricants under scuffing conditions [55], compared to bare fluids. A lower the COF means a better fluid.

**Table 3.** COF performance obtained by ITEPbP at various $SiO_2$ filler fractions.

| Oils | Pure | @0.05 wt % | @0.10 wt % | @0.15 wt % | @0.20 wt % | @0.25 wt % |
|------|------|-----------|-----------|-----------|-----------|-----------|
| COF-μ | | | | | | |
| Soybean Oil | 0.0385 ± 0.0011 | 0.0355 ± 0.0006 | 0.0344 ± 0.0006 | 0.0337 ± 0.0007 | 0.0328 ± 0.0008 | 0.0316 ± 0.0007 |
| Paraffinic Oil | 0.0429 ± 0.0005 | 0.0409 ± 0.0005 | 0.0403 ± 0.0007 | 0.0402 ± 0.0010 | 0.0395 ± 0.0007 | 0.0389 ± 0.0006 |
| Corn Oil | 0.0485 ± 0.0007 | 0.0429 ± 0.0006 | 0.0413 ± 0.0011 | 0.0409 ± 0.0005 | 0.0394 ± 0.0005 | 0.0385 ± 0.0005 |
| Sunflower Oil | 0.0437 ± 0.0007 | 0.0399 ± 0.0008 | 0.0388 ± 0.0008 | 0.0392 ± 0.0006 | 0.0384 ± 0.0007 | 0.0370 ± 0.0007 |

## 5. Conclusions

In general, all nanofluids showed a temperature-dependent behavior in thermal transport performance, indicating the role of nanoparticles oleophilic interactions. The incorporation of $SiO_2$ nanostructures within the conventional natural lubricants showed positive results overall. For the tribological performance, the load carrying capacity increased. For instance, the improvements of 45% and 60% for soybean and sunflower oils at 0.25 wt % $SiO_2$, respectively. This enhancement is due to their tribosintering onto surfaces and their spacer effect due to their small size and interlayer interaction with natural oils. These results showed the potential of $SiO_2$ nanoparticles as extreme pressure natural lubricants for metal-forming processes. The increased environmental awareness is a primary driving force for the new technological developments. Therefore, biodegradable synthetic products used in environmentally sensitive areas have great potential to succeed in industrial applications.

**Author Contributions:** Data curation, K.A.; formal analysis, J.T.-T.; investigation, J.T.-T., K.A. and J.M.D.; methodology, J.T.-T., K.A. and J.M.D.; project administration, J.T.-T; resources, J.T.-T. and J.M.D.; Supervision, J.T.-T.; writing—original draft, J.T.-T.; writing—review and editing, J.T.-T.

**Funding:** This research received no external funding.

**Acknowledgments:** Authors would like to acknowledge the support from Universidad de Monterrey.

**Conflicts of Interest:** The authors declare no conflicts of interest.

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
