# Peer review of "Tribological and Thermal Transport Performance of SiO2-Based Natural Lubricants"

_lubricants, doi:10.3390/lubricants7080071_

Round 1

Reviewer 1 Report

The authors described on the tribological and thermal transport performance of SiO2-based natural lubricants. Research on vegetable oils in terms of tribological suitability is not a novelty; only the application of SiO2 nanoparticles can be an interesting research topic. The presented results seem to be interesting, but the authors should clarify the following questions.

Suspending the particles in the oil in the mixing process, even by sonication without additives suspending and dispersing is doubtful. The section Experimental Detail is poorly written; please add more detail about the experimental procedure. The authors did not provide information about the time of sonication; and after what time from sonication, the oils were tested. Why was SiO2 selected? Will add it to the oil improve the thermal conductivity of the oil? - This is poorly explained in the text, perhaps requires in-depth research? Similarly, in terms of load capacity. Here the hardness (Mohs scale) can play an important role? No comparison with other nanomaterials such as hBN, which is definitely harder? The paragraph Introduction is too long, but the Results and Discussion section contains only 11 lines.

Author Response

Response to Reviewer 1 Comments

The authors described on the tribological and thermal transport performance of SiO2-based natural lubricants. Research on vegetable oils in terms of tribological suitability is not a novelty; only the application of SiO2 nanoparticles can be an interesting research topic. The presented results seem to be interesting, but the authors should clarify the following questions.

Point 1: Suspending the particles in the oil in the mixing process, even by sonication without additives suspending and dispersing is doubtful.

Response 1: We appreciate reviewer comments. Yes, it could seem doubtful as you mentioned. From previous work within my research group we have been exploring diverse techniques, among the best one for these types of nanofluids is ultrasonic bath dispersion, and yes, without additives also. Furthermore, a probe sonicator was also used to break any agglomeration within the nanostructures. One aspect of our research is that we want to explore the effects of not adding surfactants or additives, since these affect (in some cases, dramatically) the thermal performance of the nanofluids. These SiO2 nanostructures measure 10-20nm, which are very good for our stability purpose. We will add this information/explanation within the manuscript.

Point 2: The section Experimental Detail is poorly written; please add more detail about the experimental procedure. The authors did not provide information about the time of sonication; and after what time from sonication, the oils were tested. Why was SiO2 selected? Will add it to the oil improve the thermal conductivity of the oil? - This is poorly explained in the text, perhaps requires in-depth research?

Response 2: We appreciate reviewer comments and yes, we will add more explanation within the text of the manuscript. The sonication time and details are explained in section 2.2. Why was SiO2 selected? These nanostructures have high oleophilic behaviour and would be suitable for thermal and tribological performance of our researched conventional fluids. This incorporation is the aim of our research presented here, and the results are shown by thermal transport and tribological behaviour.

Point 3: Similarly, in terms of load capacity. Here the hardness (Mohs scale) can play an important role? No comparison with other nanomaterials such as hBN, which is definitely harder? The paragraph Introduction is too long, but the Results and Discussion section contains only 11 lines.

Response 3: We appreciate your comments. Results and discussion per subject (thermal transport and tribological) are explained in section 4. Section 5 would be a summary of our conclusions, sorry for this mistake.

Reviewer 2 Report

Suggested manuscript has an adequate literature review and introduction. Experimental par is clearly written. I would suggest to merge chapters Materials and Methods and Experimental Details because in Experimental Details are described measuring methods which can be incorporated in Materials and Methods. Results are well presented. Some of my remarks are given below.

Line 46

Additives are an alternative to increase lubricants lifetime, they can also help to increase the viscosity, helping to improve the lubricant’s properties.

Authors should explain what are additives an alternative to?

Line 136

However, the mineral oils are toxic because these content elements such S, and P, that are heavy metals, reason why they are being removed from the industry.

This sentence should be corrected because it is partially true. Sulphur and phosphorus are not heavy metals. Production of mineral oil lubricants has a very strict policy on content of phosphorus, sulphur and heavy metal continent in many of their products. This sentence should be rewritten as: However, the mineral oils can be toxic if they contain higher amounts of elements such sulphur, phosphorus and heavy metals, and that is the reason why they are being slowly replaced from the industry.

Line 144

Suggestion replace “Despite the environmental benefits of the vegetable oils, they don´t have the required properties” with Despite the environmental benefits of the vegetable oils, their propertied do not satisfy all of the required properties.

Line 153

Suggestion replace “Silica nanoparticles have been investigated as additives within lubricants, this because of the good tribological properties like anti wear and friction reduction.” With Because of the good tribological properties (anti wear and friction reduction) silica nanoparticles have been investigated as additives to lubricants.

Line 162

Authors should replace “functional groups, such as aminos and cargoxyls,” with functional groups, such as amino and carboxyl,

Line 165

Authors should replace: “are low reports” with are few reports

Line 178

Authors should replace: “soybean, sunflower and corn,” with soybean oil, sunflower oil and corn oil

Line 188

This process allows us to obtain stable nanofluids varying the filler concentration by weight, which will be further evaluated.

Authors did not test the time stability of the fluid so they should change the sentence to: This process allows us to obtain stable homogeneous nanofluids varying the filler concentration by weight, which will be further evaluated.

Author Response

Response to Reviewer 2 Comments

Suggested manuscript has an adequate literature review and introduction. Experimental par is clearly written. I would suggest to merge chapters Materials and Methods and Experimental Details because in Experimental Details are described measuring methods which can be incorporated in Materials and Methods. Results are well presented. Some of my remarks are given below.

Point 1: Line 46 - Additives are an alternative to increase lubricants lifetime, they can also help to increase the viscosity, helping to improve the lubricant’s properties. Authors should explain what are additives an alternative to?

Response 1: We appreciate reviewer comments. This was re-phrased and modified within the manuscript; additives as reinforcement of nanofluids for stability and mechanical performance. Nevertheless, the application of additives/surfactants could affect the thermal transport behaviour of nanofluids. That is one of the reasons which in our research (as well as previous research by my group) have not considered applying these additives/surfactants.

Point 2: Line 136 - However, the mineral oils are toxic because these content elements such S, and P, that are heavy metals, reason why they are being removed from the industry.

This sentence should be corrected because it is partially true. Sulphur and phosphorus are not heavy metals. Production of mineral oil lubricants has a very strict policy on content of phosphorus, sulphur and heavy metal continent in many of their products. This sentence should be rewritten as: However, the mineral oils can be toxic if they contain higher amounts of elements such sulphur, phosphorus and heavy metals, and that is the reason why they are being slowly replaced from the industry.

Response 2: We appreciate this recommendation. It has been updated within the manuscript.

Point 3: Line 144 - Suggestion replace “Despite the environmental benefits of the vegetable oils, they don´t have the required properties” with Despite the environmental benefits of the vegetable oils, their propertied do not satisfy all of the required properties.

Response 3: We appreciate this recommendation. It has been updated within the manuscript

Point 4: Line 153 - Suggestion replace “Silica nanoparticles have been investigated as additives within lubricants, this because of the good tribological properties like anti wear and friction reduction.” With Because of the good tribological properties (anti wear and friction reduction) silica nanoparticles have been investigated as additives to lubricants

Response 4: We appreciate this recommendation. It has been updated within the manuscript

Point 5: Line 162 - Authors should replace “functional groups, such as aminos and cargoxyls,” with functional groups, such as amino and carboxyl,

Response 5: We appreciate this recommendation. It has been updated within the manuscript

Point 6: Line 165 - Authors should replace: “are low reports” with are few reports

Response 6: We appreciate this recommendation. It has been updated within the manuscript

Point 7: Line 178 - Authors should replace: “soybean, sunflower and corn,” with soybean oil, sunflower oil and corn oil

Response 7: We appreciate this recommendation. It has been updated within the manuscript

Point 8: Line 188 - This process allows us to obtain stable nanofluids varying the filler concentration by weight, which will be further evaluated.

Authors did not test the time stability of the fluid so they should change the sentence to: This process allows us to obtain stable homogeneous nanofluids varying the filler concentration by weight, which will be further evaluated.

Response 8: We appreciate this recommendation. It has been updated within the manuscript

Reviewer 3 Report

The research results presented by the Authors are interesting and valuable. The presented literature is current and very wide.

However, the problem is with the graphical presentation of the test results. The graphs shown in figures 2 and 3 are difficult to compare. A better solution to this problem is to place the beginning of the Y-axis at zero points or at the another, the same value. In addition, I propose to supplement the charts with trend lines. I also suggest combining the graphs shown in Figure 3 into a single summary chart.

The text can be published in the journal "Lubricants" after correcting some graphical presentation of test results.

Author Response

Response to Reviewer 3 Comments

The research results presented by the Authors are interesting and valuable. The presented literature is current and very wide.

Point 1: However, the problem is with the graphical presentation of the test results. The graphs shown in figures 2 and 3 are difficult to compare. A better solution to this problem is to place the beginning of the Y-axis at zero points or at the another, the same value. In addition, I propose to supplement the charts with trend lines. I also suggest combining the graphs shown in Figure 3 into a single summary chart.

Response 1: We appreciate this comment. The idea for these graphs is to compare the performance of various nanofluids within conventional lubricants. Since each oil has different characteristics and behaviour on thermal / tribological. In previous research and publications, we have demonstrated similar evaluations on similar manner with very good and significant impact on the readers understanding.

For Figure 3, the tribological performance of various nanofluids under temperature-dependence evaluation is shown, if we merge or place all the nanofluids together in a summary chart it would be distorted and could have issues for readers to comprehend the chart. Another issue of the merge with the results is that they could be overlap, also being difficult for the reader to comprehend.

Round 2

Reviewer 1 Report

I recommend the article  for publication.

Author Response

We appreciate the reviewer for his/her time and effort to read, verify and review our manuscript.

Regards,

JT